# Merkel Cell Polyomavirus: Oncogenesis in a Stable Genome

**DOI:** 10.3390/v14010058

**Published:** 2021-12-30

**Authors:** Mona M. Ahmed, Camille H. Cushman, James A. DeCaprio

**Affiliations:** 1Program in Virology, Graduate School of Arts and Sciences, Harvard University, Cambridge, MA 02138, USA; ahmedm@g.harvard.edu (M.M.A.); ccushman@g.harvard.edu (C.H.C.); 2Department of Medical Oncology, Dana-Farber Cancer Institute, Boston, MA 02215, USA; 3Department of Medicine, Brigham and Women’s Hospital, Harvard Medical School, Boston, MA 02115, USA

**Keywords:** Merkel cell polyomavirus, Merkel cell carcinoma, large T, small T, cancer hallmarks, genomic instability

## Abstract

Merkel cell polyomavirus (MCV) is the causative agent for the majority of Merkel cell carcinoma (MCC) cases. Polyomavirus-associated MCC (MCCP) is characterized by the integration of MCV DNA into the tumor genome and a low tumor mutational burden. In contrast, nonviral MCC (MCCN) is characterized by a high tumor mutational burden induced by UV damage. Since the discovery of MCV, much work in the field has focused on understanding the molecular mechanisms of oncogenesis driven by the MCV tumor (T) antigens. Here, we review our current understanding of how the activities of large T (LT) and small T (ST) promote MCC oncogenesis in the absence of genomic instability. We highlight how both LT and ST inhibit tumor suppressors to evade growth suppression, an important cancer hallmark. We discuss ST interactions with cellular proteins, with an emphasis on those that contribute to sustaining proliferative signaling. Finally, we examine active areas of research into open questions in the field, including the origin of MCC and mechanisms of viral integration.

## 1. Introduction

The hallmarks of cancer have been refined and expanded since the original landmark publication “Hallmarks of Cancer” by Hanahan and Weinberg as our understanding of the requirements for cancer initiation and progression has improved [1,2,3]. In the 2011 follow-up, called Hallmarks II, genomic instability was identified as an enabling trait that supported the development of cancer hallmarks [2]. Genomic instability can be defined as “higher than normal rates of mutation”, and it was proposed that an increased mutation rate would lead to cellular transformation when mutations arise in proteins involved in the DNA damage response (reviewed in [4]). Mutations in the Hallmark pathways are thought of as “driver” mutations, but genomic instability can also lead to “passenger” mutations, or mutations that contribute little to no effect on cancer progression. Identifying driver mutations can therefore be challenging, especially in tumors with a high mutational burden.

The study of polyomaviruses has played an important role in identifying the pathways that when disrupted promote cancer development. In particular, the study of the simian virus 40 (SV40) led to the identification of a set of minimal mutations required for cellular transformation [5,6,7], as well as the discovery of p53 [8,9] while the study of murine polyomavirus led to the discovery of PI3K signaling [10]. SV40 is well documented to induce tumors in rodents such as hamsters injected with the virus, but there is controversy whether SV40 can contribute to the development of human malignancies as well (reviewed in [11]). The human polyomavirus Merkel cell polyomavirus (abbreviated MCV or MCPyV) is unequivocally linked to Merkel cell carcinoma (MCC), an aggressive neuroendocrine carcinoma of the skin [12,13,14]. Interestingly, the discovery of MCV in MCC revealed a dichotomy between MCC tumors of viral origin and those of non-viral origin. Nonviral MCC (MCCN) is characterized by a high tumor mutational burden while polyomavirus-associated MCC (MCCP) has a low tumor mutational burden with strikingly few genomic aberrations [15,16]. Although MCV proteins have been associated with increased genomic instability, the noticeable lack of mutations in MCCP cancers suggests that their genomes are relatively stable. In contrast, MCCN tumors are clearly driven by mutations, including many that contribute to genomic instability. Despite the striking difference in etiology, MCCP and MCCN tumors are clinically and phenotypically very similar to each other [17]. In this review, we will discuss how the transforming activities of the MCV viral proteins in a relatively stable genomic background in MCCP can mimic genomic instability and the high mutational burden observed in MCCN.

## 2. Two Types of Merkel Cell Carcinoma

MCC is a rare skin cancer that typically presents on sun-exposed areas of the skin in the elderly and fair-skinned although it can present even more rarely in children and darker skin individuals [17]. The incidence of MCC in the U.S. is low, 0.7 per 100,000 person-years in 2013, but has risen by 95% between 2000 and 2013 and is predicted to continue to rise due to a growing elderly population [18]. Immunosuppression is also a risk factor for developing MCC, suggesting that this cancer may have an infectious origin [19]. Indeed, recognition of this risk factor led to the discovery in 2008 of a previously unidentified polyomavirus clonally integrated in 8 out of 10 MCC tumors studied [12]. This novel polyomavirus was named Merkel cell polyomavirus and paved the way to our current understanding that there are two distinct etiologies of MCC: nonviral and polyomavirus-associated MCC.

MCCN and MCCP tumors present similarly as a rapidly growing, painless, red, or skin-colored lesions typically in the dermal layer of the skin [17]. Both MCCN and MCCP have a high grade neuroendocrine histology with immunohistochemical staining with cytokeratin 20 (CK20) and other skin keratinocytes as well as neuroendocrine markers, such as synaptophysin (SYP), chromogranin A (CHGA), and INSM1 [20]; reviewed in [21]. MCCN tumors can occasionally be distinguished from MCCP tumors, by increased nuclear polymorphism and abundant cytoplasm [22,23], as well as staining for CK7 and thyroid transcription factor-1 (TTF1) and lack of neurofilament expression [22,24]. MCCN tends to be the more aggressive subtype of MCC and is associated with an increased risk of cancer progression, recurrence, and death [25].

Despite their similar gross and histologic appearance, MCCN and MCCP tumors differ greatly from each other on a genetic level. MCCN tumors are characterized by a very high tumor mutational burden often greater than 20 mutations/Mb [15,16]. MCCN tumors have a UV mutational signature (Signature 7), defined by a predominance of C > T mutations at dipyrimidine sites, reflecting an accumulation of UV-induced damage from excessive sunlight exposure [15,26]. The most commonly mutated genes in MCCN tumors are the *RB1* and *TP53* tumor suppressor genes, *NOTCH1*, *PIK3CA*, and *KMT2D* [27]. Additionally, MCCN tumors also frequently contain an amplification of the *MYC* homolog, *MYCL* [28]. In contrast, MCCP tumors are characterized by a low tumor mutational burden and usually contain wild type *RB1* and *TP53* [27]. However, when mutations are detected in MCCP tumors, they are most frequently found in the *RB1*, *TP53*, and *PTEN* genes. Although more common in MCCN tumors, biallelic deletion of *RB1* or *TP53* or both has also been observed in MCCP tumors [28]. MCCP tumors have a slight enrichment of C > T and T > C mutations on the non-transcribed strand [15]. The mutational profile of MCCP tumors most closely resembles COSMIC mutational signature 5, a nonspecific signature with no known etiology that has been observed in other cancer types [15,26,27].

Oncogenesis in MCCP is driven largely by viral protein activity that perturbs at least some of the same signaling pathways that are affected by mutations in MCCN. For example, *MYCL* is amplified in MCCN and activated by viral proteins in MCCP [28,29]. Similarly, while MCCN tumors contain inactivating mutations in *RB1* and *TP53*, viral proteins functionally inactivate RB and p53 in MCCP [27,30,31]. Interestingly, the combined inactivation of *RB1* and *TP53* was required to reprogram normal prostate epithelial cells into small cell prostate cancer, another neuroendocrine cancer [30]. The activities of the MCV viral proteins are discussed in greater detail below.

## 3. Origins of MCC

The cell of origin for the two forms of MCC has been widely debated and remains an active area of research. MCC was first identified as a trabecular carcinoma of the skin by Cyril Toker in 1972 [31]. The similarity in morphology between MCC and Merkel cells, a neuroendocrine cell found in human hair follicles, was noted by Toker and others [32,33]. In addition to morphology, MCC cells also share some markers with Merkel cells such as CK20 [34,35], CD56 [36,37,38], synaptophysin [36,39,40,41], and neurofilament expression [41] and this resemblance is how MCC gets its name [37,42]. However, despite the similarities in appearance, it is not clear whether MCC tumors are derived from normal Merkel cells since their location is within the epidermis while MCCs are typically found in the dermis, the divergent patterns of CK20 staining, the presence of additional markers not found in Merkel cells, and the lack of proliferating Merkel cells or the detection of any benign Merkel cell tumors (reviewed in [43,44]).

A number of alternative cell types have been proposed to be the cell type of origin for MCC, including epidermal stem cells [44,45], dermal stem cells [46], and pre/pro-B cells [47] (reviewed in [46,48,49]). The evidence for one cell type over another has been driven largely by expression patterns of different cell type-specific markers and the location of MCCs in the layers of the skin. Normal Merkel cells are derived from epidermal stem cells [48,50,51] and there have been reported cases of collision tumors between MCC and keratinocytic neoplasms in the epidermis [49,52,53,54] supporting the epidermal stem cell origin theory. Genomic evidence has linked both MCCN tumors [53,54] and MCCP tumors [52] with adjacent epithelial tumors suggesting that both forms of MCC can derive from keratinocytic tumors. Additionally, the expression of CK14 and CK19, two markers associated with epidermal stem cells [55,56], have been identified in a set of MCCs. However, many MCCs are found in the dermal or subcutis layers of the skin [57], suggesting that they may arise from the dermal layer. Dermal stem cells derive from the neural crest lineage [58,59], and a neural crest origin could explain the expression of neuronal cell markers in MCC like neurofilament, neuron-specific enolase, and synaptophysin [46]. SOX2, a neural crest-derived stem cell factor [60], is also detected in MCCs [61]. In addition to epidermal and dermal stem cell markers, the expression of B-lymphoid lineage markers in MCC has been reported. PAX5, also known as B-cell specific activator protein (BSAP), is critical for B-cell commitment and is often found in MCCs [47,62,63,64]. Tdt, C-KIT, and SCF are additional markers found in MCC that play important roles in B-cell development, although the expression levels of each in MCC can vary based on the study [62,65,66,67]: reviewed in [68].

The differences in marker expression, localization of tumors, and the striking differences in mutation burden among MCC tumors have led some to suggest that there are different cell types of origin for MCCP and MCCN tumors. Sunshine et al. posit that MCCN tumors may arise from the epidermis while MCCP tumors arise from the dermis [69]. Indeed, the lack of a UV-mutational signature in MCCP tumors suggests they arise from a sun-protected cell type whereas the strong UV-mutational signature in MCCN tumors reflects UV radiation damage from excessive, lifelong exposure to sunlight [15,16,70,71]. One strategy to clarify the cell type of origin for MCCP has been to identify cell types permissible to MCV infection. In a 2016 study, Jianxin You and colleagues found that human dermal fibroblasts can support MCV infection [72]. Dermal fibroblasts and dermal stem cells are more protected from UV-irradiation than cells of the epidermis, so this theory would explain the lack of a UV-mutation signature in MCCP cells. Identifying cell types permissible to MCV infection is a promising line of investigation to not only determine the cell type of origin of MCC but also to gain a better understanding of the molecular mechanisms of MCCP development. In the following sections, we review the literature on MCV biology and potential mechanisms of viral-mediated oncogenesis.

## 4. Merkel Cell Polyomavirus

All polyomaviruses, including MCV, are non-enveloped, double-stranded DNA viruses with a circular genome encoding early and late gene regions on opposite strands with a non-coding control region (NCCR) located between them that contains the viral origin of replication [12]; reviewed in [70,73] (Figure 1A). During viral infection, the early gene region is expressed immediately and before viral DNA replication begins. The early region undergoes alternative splicing to produce two T antigens. Both T antigens contain an N-terminal J domain that binds to cellular heat shock proteins. Large T (LT) is required for viral DNA replication and contains an RB-binding motif (LXCXE), an origin-binding domain (OBD), a nuclear localization signal, and a helicase domain (Figure 1B). Small T (ST) is expressed from transcriptional read-through of the splice site used by LT and therefore shares only the J domain with LT and encodes a unique region that plays an important role in mediating ST-host protein interactions. An alternatively spliced form of LT, 57 kT, consists of the J domain, the RB-binding domain, some of the OBD, and the C-terminal 100 amino acid residues of LT (Figure 1A). The early region also encodes for a third protein termed ALTO from an alternative open reading frame in the second exon of LT [17]. Viral DNA replication is driven by LT and enhanced by ST. The functions of 57 kT and ALTO are unknown.

The late gene region of MCV encodes for the capsid proteins, viral protein 1 (VP1) and viral protein 2 (VP2) (Figure 1A). These structural proteins are expressed after viral DNA replication has been initiated [70]. In addition, the late region also encodes a viral microRNA that regulates early gene expression [71]. This miRNA, a 22-nucleotide sequence, is complementary to *LT* just downstream from the region encoding the RB-binding motif (nucleotides 1217–1238) [74]. The MCV miRNA is expressed at low levels in about half of MCCP tumors and its contribution to MCC tumorigenesis is unclear.

MCV is a highly prevalent virus that causes a persistent, lifelong, and generally innocuous infection in most people [17]. Antibodies to MCV VP1 are detected in the sera of most healthy adults and children, indicating that infection with MCV likely occurs in early childhood and persists throughout adult life [75]. While VP1 antibodies are also detected in MCC patients, their levels do not correlate with disease progression and are not a useful clinical biomarker [76]. However, MCCP patients may have higher titers of VP1 antibodies than healthy individuals [77]. In contrast, healthy individuals rarely have antibodies to the T antigens but a majority of MCCP patients have detectable levels of serum antibodies to LT and ST [76]. T antigen antibody titers are a reliable biomarker to monitor cancer remission and recurrence. Following treatment and during cancer remission, T antigen antibody titers dramatically decrease and remain at low levels. An increase in antibody titers following treatment suggests cancer recurrence and prompts a search for disease.

MCCP tumors constitutively express wild-type ST and a truncated form of LT that eliminates the origin-binding domain, the helicase domain, and in some cases, the nuclear localization signal (Figure 1B) [17]. The LT truncation mutations eliminates the ability of the integrated virus to replicate. MCV is often integrated into host DNA as head-to-tail concatemers [12,78]. The mechanism of MCV integration into the genome is poorly understood and multiple groups have attempted to understand this process through sequencing of MCC cell lines and tumors. To date, all integration events have been observed at unique sites in the genome, indicating that there are no integration “hot spots” [15,28,79]. Integration events are frequently observed either as a single integration or as two integrations separated by host DNA. In addition, host DNA surrounding the integrant often appears to have been amplified. Microhomology of 4–7 base pair sequences between the human and MCV genomes have been observed at integration sites [28]. Two groups have independently proposed microhomology-mediated end joining as the mechanism of MCV integration. Starrett et al. proposed that linearized MCV DNA anneals to resected host DNA at the site of a double-strand break (Figure 2) [15,28]. The viral origin of replication is activated, perhaps by expression of full-length wild type LT expressed from intact viral genomes and leads to rolling circle amplification of viral DNA and the surrounding host DNA. This forms a circular virus–host DNA intermediate that is repaired and reintegrated by the host DNA repair machinery, resulting in amplified host DNA flanked by viral DNA. Czech-Sioli et al. proposed that both host and linear viral DNA undergo end resection [80]. Viral DNA is annealed to host DNA at microhomology sequences. This model proposes that the observed amplification of host DNA is a result of DNA synthesis via microhomology-mediated break-induced replication following invasion of the annealed viral DNA into a homologous region of host DNA. Furthermore, Czech-Sioli et al. proposed that nonhomologous end joining is the mechanism of MCV integrations that appear as a single integration event. Both models agree that the MCV genome is linearized prior to integration and that this linearization most likely occurs during errors in viral DNA replication [15,28,79]. Tumors containing viral concatemeric integrants display identical LT mutations on each viral genome, suggesting that LT truncation occurs prior to viral integration [28,79,81].

## 5. LT Inhibits RB

Although MCCP tumors express a highly mutated and truncated form of LT, it retains the RB binding LXCXE motif indicating that this is an essential oncogenic activity for MCV LT [82]. LT directly binds to and inhibits the retinoblastoma tumor suppressor protein, RB [83]. RB is a transcriptional repressor that acts as the “guardian” of the G1/S cell cycle checkpoint [81]. During this cell cycle checkpoint, the cell evaluates intracellular and extracellular conditions and makes the decision to progress through the cell cycle into S phase or remain in G1 and enter quiescence. RB inactivation is the key molecular event required for entry into S phase. In the G0 and early G1 phases of the cell cycle, RB binds and inhibits the E2F transcription factor [79]. The RB-E2F interaction prevents expression of cell cycle genes required for entry into S phase. During the G1-S transition in normal cells, RB phosphorylation by the cyclin dependent kinases releases E2F, thereby allowing for expression of E2F-dependent cell cycle genes and S phase entry. Once a cell has passed through the G1/S checkpoint, it is committed to progressing through the cell cycle independently of extracellular growth signals [84,85].

Inhibition of RB by LT allows for the inappropriate activation of E2F and unregulated cell cycle progression. Knockdown of T antigen expression in MCC cells inhibited cell growth which could not be rescued by expression of a mutant LT protein incapable of RB binding, demonstrating that the LT-RB interaction is essential for the growth of MCC cells [14]. The inhibition of RB by SV40 LT requires both the LXCXE motif and the N-terminal J domain, which binds to the heat shock protein, HSC70, and stimulates its ATPase activity [86,87,88,89,90,91]. A point mutation in MCV LT that prevents binding to HSC70 inhibits proliferation of MCC cells and reduces expression of E2F target genes, suggesting that MCV LT may inhibit RB through its interaction with HSC70 in a manner similar to SV40 LT [92].

Loss of RB function was first discovered in retinoblastoma tumors and has since been observed in a wide range of human cancers [93,94]. Additionally, many viral oncoproteins bind and inactivate RB, including SV40 LT and human papillomavirus E7 (reviewed in [95]). The inactivation of RB by LT allows MCCP cancer cells to avoid growth suppression, a hallmark of cancer [2]. The importance of RB binding for MCCP oncogenesis is evidenced by the fact that the RB binding site is always preserved despite the numerous LT mutations found in MCCP tumors [82]. The observation that an intact ST was also always expressed in its wild-type form in MCCP, coupled with the observation that MCV LT did not bind to p53, suggested that ST may be playing a crucial role in promoting MCCP oncogenesis and acquisition of the cancer hallmarks [28,96].

## 6. MCV ST Interactions

MCV ST is sufficient to induce cellular transformation in Rat1 fibroblast cells [97] and tumor formation in p53-null transgenic mice [98], suggesting that ST is a major driver of oncogenesis. To understand the activities of MCV ST that drive tumor development, many molecular studies in the field have worked to identify the binding partners of ST and characterize the significance of these interactions. Here, we summarize these findings for three major binding partners.

### 6.1. Protein Phosphatase 2A (PP2A)

PP2A refers to a family of holoenzymes with Ser/Thr phosphatase activity composed of at least three components; a structural subunit (PP2A A), a catalytic subunit (PP2A C), and a variable subunit (PP2A B) that distinguishes PP2A binding partners (reviewed in [99]). There are two distinct PP2A A genes (α and β), two PP2A C genes (α and β), and a multitude of different PP2A B subunits that have been categorized into 4 families: B, B’, B’’, and B’’’. Many polyomavirus ST proteins have been shown to displace specific PP2A B subunits including SV40, JCPyV, BKPyV, HPyV6, TSPyV, and MCV (reviewed in [100,101]). SV40 ST can bind to PP2A Aα subunits and prevent the assembly of PP2A complexes containing B55α (B), B56α (B’), or B56ε (B’) subunits. More recently, SV40 ST was also shown to bind to striatin proteins (B’’’) and promote PP2A-mediated MAP4K4 dephosphorylation [102]. In contrast, MCV ST binds to PP2A Aα and Aβ and has only been shown to inhibit complex assembly with the B56α subunit [103]. The residues for PP2A binding have been mapped for MCV ST to 4 critical residues [103] (Figure 3A,B). PP2A binding is thought to be critical for the oncogenic activities of SV40 ST [6,102] but is dispensable for MCV ST-mediated cellular transformation [97]. In accordance with this understanding, SV40 ST prevents PP2A-mediated dephosphorylation of AKT while MCV ST does not [104]. Additionally, loss of PP2A binding does not affect the ability of MCV ST to promote LT-mediated viral DNA synthesis [104]. From these findings, we conclude that the critical oncogenic activities of MCV ST lie with alternative protein interactions.

### 6.2. SKP1-CUL1-FBOX (SCF) Complexes

MCV ST may contribute to viral replication and oncogenesis by stabilizing MCV LT expression, a phenomenon that has been documented by multiple groups [104,109]. Kwun et al. suggest that ST stabilizes LT by binding and inhibiting the SCF(FBW7) ubiquitin ligase complex, which they found targets LT for degradation (Figure 3C). The SKP1, CUL1, and F-box complex (SCF) is a multiprotein cullin-RING ubiquitin ligase with an F-box substrate recognition component. FBW7 (also known as FBXW7, CDC4, AGO, and SEL10) is widely regarded as a tumor suppressor, mainly for its role in mediating degradation of oncoproteins such as c-MYC, Cyclin E, and NOTCH (reviewed in [110]). Kwun et al. identified ST amino acids 91–95 as critical for the LT stabilization domain (LSD) [104] (Figure 3A). They went on to show that substituting the LSD residues with alanines greatly reduced ST-mediated transformation of Rat-1 cells [104], suggesting that this region of ST plays a critical role in its oncogenic activities. The LSD domain has also been linked to additional oncogenic activities of ST, including activation of non-canonical NF-kB signaling [111], and upregulation of matrix metalloproteinase expression [112].

In a follow up paper, Kwun et al. found that wild-type (WT) ST expression induced genomic instability that was attenuated by mutations in the LSD, suggesting that this region of ST is crucial for its oncogenic activities and may in fact contribute to some genomic instability despite the low mutation burden seen in MCCP cells [113]. In addition to SCF(FBW7), Kwun et al. also found that ST binds to SCF(B-TRCP) in an LSD domain-specific manner (Figure 3C) and show that the expression of targets of SCF(FBW7) and SCF(B-TRCP) such as c-MYC, Cyclin E, and PLK2 are stabilized with WT ST expression. Notably, the direct interaction between SCF(FBW7) and the T antigens has been called into question. Dye et al. reported that neither MCV ST nor MCV LT contain the canonical FBW7 phospho-degron, and they were unable to reproduce the finding that MCV ST binds to either SCF(FBW7) or SCF(B-TRCP) [109]. This study did support the finding that the LSD domain of ST stabilizes LT, suggesting that ST may promote LT expression through an alternative mechanism. Further work is necessary to explain the discrepancy between these findings and to clarify the function of the LSD domain.

### 6.3. ST, MYCL, and P400: The SLaP Complex

To identify additional binding partners of ST, Cheng et al. performed an immunoprecipitation of MCV T antigens and identified associated proteins by mass spectrometry [29]. This study found that MCV ST uniquely binds to the oncogenic transcription factor L-MYC (MYCL) and its heterodimerization partner MAX along with all the components of the transcriptional regulatory complex known as the TIP60 (KAT5)/P400 (EP400) complex (Figure 3D). Here, we will refer to the ST, MYCL, and P400 complex as the SLaP complex. ST binding to MYCL/MAX and the TIP60/P400 complex required two glutamic acid residues (E86, E87) (Figure 3A). These residues are predicted to be located proximal to the LSD domain. Notably, Cheng et al. observed that substitution or deletion of the LSD also reduced SLaP complex formation. ChIP-seq and RNA-seq experiments performed with MCCP cell lines revealed that genes directly bound by the SLaP complex at their promoters had reduced expression following knockdown of P400, MYCL, or ST/LT [29]. These data suggest that the SLaP complex acts as a transcriptional activator, promoting the transcription of a subset of genes.

Further investigation into the downstream gene targets of the SLaP complex has provided additional insights into the biology of MCCP. Two targets of the SLaP complex are MDM2 and CK1α (CSNK1A1), proteins that promote degradation of p53 through ubiquitin targeted proteasomal degradation. Based on this finding, Park et al. demonstrated that MDM2 and CK1α, in complex with MDM4, promote p53 degradation in MCCP cells while MDM2 and MDM4 inhibitors stabilized p53 and promoted p53-mediated tumor cell death [114]. This finding shed light on a previously described characteristic of MCC tumors; the majority of MCCP tumors contain wild type p53 genes, whereas p53 is frequently mutated in MCCN tumors [27]. This work suggested that MCV may indirectly reduce p53 activation by activating MDM2 and CK1α expression to promote p53 degradation. Furthermore, they observed that MCV LT does not bind and inhibit p53; instead, they found that LT activates p53 [96,114]. The observation that MCCP tumors are sensitive to MDM2/MDM4 inhibition is being investigated in clinical trials. Inhibition of p53 by ST may at least partially explain how ST expression promotes genomic instability as previously reported [113], but this hypothesis has yet to be formally tested.

MDM2 and CK1α are not the only targets of SLaP that play a critical role in MCCP cell survival. Multiple components of the LSD1/CoREST transcriptional repressor complex are transcriptionally activated by SLaP, and MCCP cells were shown to be sensitive to LSD1 inhibition [115]. Similar to normal Merkel cells, MCC tumor cells express ATOH1, the transcription factor driving Merkel cell differentiation. Park et al. demonstrated that the LSD1 repressor complex binds to several ATOH1 targets and reduced transcription of differentiation genes that would otherwise prevent tumor growth [115].

Although significant progress has been made into understanding the molecular mechanisms of ST-mediated oncogenesis since the discovery of MCV in 2008, there remain many unanswered questions. Conflicting reports regarding the interaction of MCV ST with SCF complexes coupled with the finding that mutating the LSD domain reduces SLaP complex formation suggests that transcriptional regulation by a ST-MYCL complex may play a larger role in oncogenesis than previously appreciated.

## 7. Conclusions

Our understanding of MCC has provided great insight into how viral oncoproteins mimic the effect of thousands of mutations in nonviral cancers. In this review, we have described how two viral proteins, LT and ST, disrupt specific signaling pathways that control important cellular processes that contribute to oncogenesis yet maintain genomic integrity. LT and ST cooperate to acquire the biological capabilities known as the hallmarks of cancer to promote the development of MCCP [1,2]. Both LT and ST evade growth suppressors through the inhibition of RB and p53, respectively [83,114]. ST sustains proliferative signaling by activating MYCL [29]. Future work is needed to understand how LT and ST contribute to other hallmarks, such as inducing angiogenesis, as well as emerging hallmarks, such as avoiding immune destruction. The study of MCC provides a unique model system to gain insights into neuroendocrine and viral cancers.

## Figures and Tables

**Figure 1 viruses-14-00058-f001:**
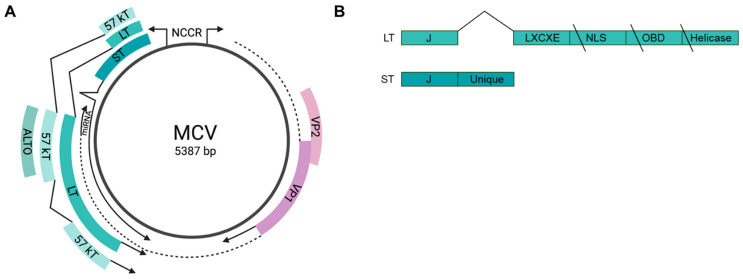
MCV genome. (**A**) Circular map of the MCV genome depicting early (green) and late (purple) genes and the non-coding control region (NCCR). Created with BioRender.com (accessed on 1 November 2021) (**B**) The early gene region undergoes alternative splicing to produce LT and ST. The C-terminus of LT is truncated by mutations in MCC, eliminating the nuclear localization signal (NLS), origin binding domain (OBD), and helicase domains while preserving the LXCXE RB-binding motif.

**Figure 2 viruses-14-00058-f002:**
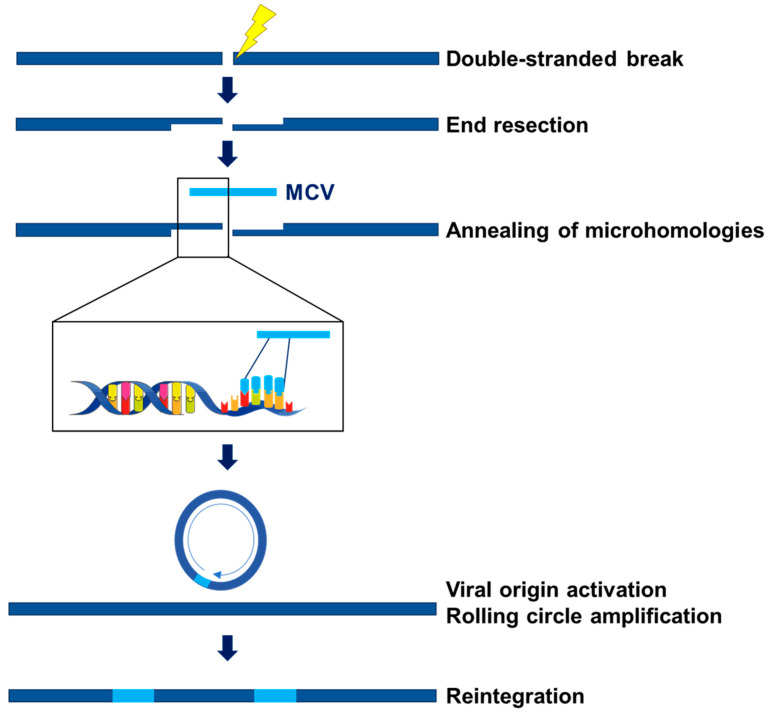
Proposed model of MCV integration as described by Starrett et al. [15,28]. A double-stranded break in the host genome activates the DNA damage response. The DSB ends of the host DNA undergo end resection, creating single-stranded DNA overhangs. A linearized MCV genome anneals to resected host DNA at microhomologous sequences. The viral origin of replication is activated, leading to rolling circle amplification of viral DNA and the surrounding host DNA. DNA repair mechanisms facilitate the resolution and reintegration of the circular virus–host intermediate, resulting in linear cellular DNA flanked by viral DNA. Part of this image was modified from smart.servier.com, accessed on 1 November 2021.

**Figure 3 viruses-14-00058-f003:**
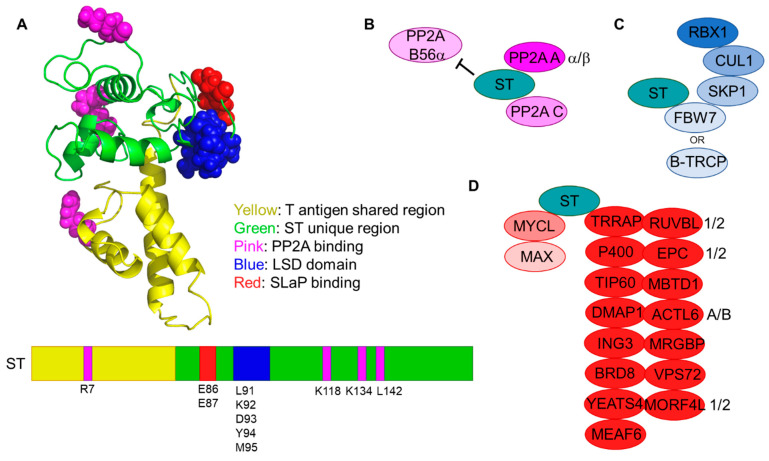
The domains and binding partners of MCV ST. (**A**) The structure of MCV ST was modeled by threading the MCV ST sequence onto the SV40 ST crystal structure (PDB ID: 2PF4) using the I-TASSER server [105,106,107,108] and regions of interest were highlighted. (**B–D**) Cartoons of the major ST-containing cellular complexes: (**B**) PP2A; (**C**) SCF(FBW7) and SCF(B-TRCP); (**D**) SLaP: ST, MYCL, and P400 complex.

## Data Availability

Not applicable.

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
