# Peer review of "Merkel Cell Polyomavirus: Oncogenesis in a Stable Genome"

_viruses, 2021, doi:10.3390/v14010058_

Round 1
Reviewer 1 Report
This is a very well written review with updated information.
I would like to suggest one addition to page 6 lines 271-272.
HPyV sT- PP2A interaction was found for HPyV6 and TSPyV sT as well. Both sT diplayed binding to PP2A-A and PP2A-C subunits when overexpressed in HEK293 cells.
(reviewed by Moens U, Macdonald A. Effect of the Large and Small T-Antigens of Human Polyomaviruses on Signaling Pathways. Int J Mol Sci. 2019 Aug 12;20(16):3914. doi: 10.3390/ijms20163914. PMID: 31408949; PMCID: PMC6720190.)
Author Response
Thank you for the opportunity to submit a revised draft of our manuscript, “Merkel cell polyomavirus: Oncogenesis in a stable genome”. We are grateful for the insightful feedback from the reviewers and have incorporated their suggestions in our revised draft. The following changes have been made to the manuscript:
As suggested by the reviewer, we added HPyV6 and TSPyV as examples of polyomaviruses whose ST proteins interact with PP2A and cited PMID: 31408949.
Reviewer 2 Report
We thanks the authors and editors to allow us to read this great review produced by an expert team in the field. I strongly recommand its publication
I only have few comments :
1)Page 2 : the authors stated that MCCP and MCCN tumors are clinically and phenotypically very similar. However there is an increasing amount of evidence in the litterature that MCCN cases harbor worst outcome than the others. Moreover several morphologic and imlmunohistochemical features distinguishing both groups have been described, please highlight these findings
References :
PMID: 27815175, PMID: 33547200, PMID: 23664542, PMID: 30067951, PMID: 31201352
2) Page 2 : SOX2 is cited as a neuroendocrine marker however to my konwledge it is not used in current practise and SOX2 is mentionned neither in ref 20 or 21
3) Page 2 : the authors wrote that TP53 and RB1 mutations are also the most commeon mutations in MCCP cases, I would be curious to know whether biallelic inactivation of the genes can be observed in this setting
4) Page 3 : regarding the hypothesis that MCC might derive from epidermal stem cells, a better reference shoudl be used. Indeed reference 42 investigated the presence of “cancer stem cells” in MCC which is to my view a different question than the identification of the cell of origin
May be the following reference would be more appropriate :
PMID: 23304516,
5) Page 3: Simillarly regarding combined tumors, at least three recents studies investigated the genetic backgrounds of these tumors and demonstrated the clonal link between the two components, please add these references
PMID: 34593967, PMID: 34480892, PMID: 31759946
6) Page 6: regarding Large T KD by Houben et al. , to my view this experiment resulted in reduced tumor growth rather than cell death as stated in the manuscript
7) Page 6 : regarding RB1 inactivation in cancer, I think it would be interesting to mention that copbined TP53 and RB1 inactivations is an genetic hallmark of neuroendocrine carcinomas (PMID: 30287662)
Author Response
Thank you for the opportunity to submit a revised draft of our manuscript, “Merkel cell polyomavirus: Oncogenesis in a stable genome”. We are grateful for the insightful feedback from the reviewers and have incorporated their suggestions in our revised draft. The following changes have been made to the manuscript:
We highlighted the morphological and immunohistochemical features that distinguish MCCN tumors from MCCP tumors. We also noted that MCCN is the more aggressive subtype of MCC. As suggested by the reviewer, we cited the following references: PMID: 27815175, PMID: 23664542, PMID: 30067951, PMID: 31201352.
We removed the mention of SOX2 as a neuroendocrine marker.
We highlighted the evidence of MCCP tumors with biallelic RB1 or TP53 deletions from PMID: 32188490.
As suggested by the reviewer, we cited PMID: 23304516 to support the hypothesis that MCC is derived from epidermal stem cells.
We cited the following additional references suggested by the reviewer to support the hypothesis that MCC is derived from keratinocytic tumors: PMID: 34593967, PMID: 34480892, PMID: 31759946
We corrected a mistake identified by the reviewer and noted that T antigen knockdown inhibited cell growth in MCCP cells. The original manuscript mistakenly stated that T antigen knockdown induced cell death in MCCP cells.
As suggested by the reviewer, we included the finding from PMID: 30287662 that RB and p53 inactivation were required to reprogram prostate epithelial cells into small cell prostate cancer, another neuroendocrine cancer.